# Structural and Immunoreactivity Properties of the SARS-CoV-2 Spike Protein upon the Development of an Inactivated Vaccine

**DOI:** 10.3390/v15020480

**Published:** 2023-02-09

**Authors:** Larisa V. Kordyukova, Andrey V. Moiseenko, Marina V. Serebryakova, Marina A. Shuklina, Maria V. Sergeeva, Dmitry A. Lioznov, Andrei V. Shanko

**Affiliations:** 1Belozersky Institute of Physico-Chemical Biology, Lomonosov Moscow State University, 119991 Moscow, Russia; 2Faculty of Biology, Lomonosov Moscow State University, 119991 Moscow, Russia; 3WHO National Influenza Center, Smorodintsev Research Institute of Influenza, 197376 Saint-Petersburg, Russia; 4R&D Department, FORT LLC, 119435 Moscow, Russia

**Keywords:** SARS-CoV-2 inactivation, β-propiolactone, formaldehyde, UV irradiation, S protein, pre-fusion, post-fusion, transmission electron microscopy, MALDI-TOF mass spectrometry, ELISA, immunoreactivity

## Abstract

Inactivated vaccines are promising tools for tackling the COVID-19 pandemic. We applied several protocols for SARS-CoV-2 inactivation (by β-propiolactone, formaldehyde, and UV radiation) and examined the morphology of viral spikes, protein composition of the preparations, and their immunoreactivity in ELISA using two panels of sera collected from convalescents and people vaccinated by Sputnik V. Transmission electron microscopy (TEM) allowed us to distinguish wider flail-like spikes (supposedly the S-protein’s pre-fusion conformation) from narrower needle-like ones (the post-fusion state). While the flails were present in all preparations studied, the needles were highly abundant in the β-propiolactone-inactivated samples only. Structural proteins S, N, and M of SARS-CoV-2 were detected via mass spectrometry. Formaldehyde and UV-inactivated samples demonstrated the highest affinity/immunoreactivity against the convalescent sera, while β-propiolactone (1:2000, 36 h) and UV-inactivated ones were more active against the sera of people vaccinated with Sputnik V. A higher concentration of β-propiolactone (1:1000, 2 h) led to a loss of antigenic affinity for both serum panels. Thus, although we did not analyze native SARS-CoV-2 for biosafety reasons, our comparative approach helped to exclude some destructive inactivation conditions and select suitable variants for future animal research. We believe that TEM is a valuable tool for inactivated COVID-19 vaccine quality control during the downstream manufacturing process.

## 1. Introduction

In vaccine prophylaxis, virus inactivation technologies have been used for a long time [1]. This type of vaccine, in addition to the main surface antigens, also contains other structural proteins, the presence of which might improve immune response [2]. It has several advantages over other types of vaccines, both at the development and production stages, especially in low-income and developing countries. In the global fight against the COVID-19 pandemic, roughly estimated, at least half of all vaccines used in the world are of the inactivated type [3,4,5]; these are the CoronaVac and Sinopharm BIBP COVID-19 vaccines, which are widely used in Southeast Asia and the countries of Latin America, and the Covivac vaccine in Russia. However, according to numerous studies, inactivated vaccines, compared to genetically engineered ones based on viral vector or mRNA technologies, showed relatively lower protection rates [6,7,8,9]. Due to the huge volume of production of inactivated vaccines in the world and their promising role in the eradication of the pandemic, it is necessary to evaluate many different factors that may affect the safety of vaccines, their effectiveness, and the quality of immune protection.

The main antigen of SARS-CoV-2, the infectious agent of COVID-19, is the S-protein which forms homotrimeric spikes on the virion’s surface [10,11,12,13]. Cryo-EM and cryo-ET analysis make it possible to distinguish two main conformations of the S spike: a pre-fusion S1/S2 trimer, which dominates before the virus interacts with the ACE2 receptor on the cell surface, and a post-fusion S2 trimer, exposed after the dissociation of the S1 subunits from the spike provoked by their interaction with the receptor [14]. During virus entry, a change in the conformation of S-trimers triggers the fusion of the viral and cellular membranes, which leads to the entrance of the viral genomic RNA into the cell [15].

Various virus-inactivating agents applied during inactivated vaccine manufacturing, such as formaldehyde, β-propiolactone (BPL), or UV radiation [16,17,18,19,20], may affect the S-protein’s conformation status. For example, according to Ke et al. [10], approximately 97% of the S-spikes on the surface of formaldehyde-fixed virions were in the pre-fusion conformation, and 3% of the S-spikes were in the post-fusion conformation. When virions were inactivated by BPL, the situation was completely different; the proportion of spikes in the post-fusion conformation was up to 76% versus 24% in the pre-fusion conformation, according to some studies [11]. Nevertheless, BPL began to be tested as an inactivating reagent for the production of vaccines against COVID-19 [21,22,23,24,25].

SARS-CoV-2 virions bearing on their surface a large number of spikes in post-fusion conformation are significantly weakened in terms of their penetration into cells. Being part of inactivated vaccines, such virions are likely to elicit the production of so-called non-neutralizing or sub-neutralizing antibodies that are unable to neutralize a live virus. In the worst-case scenario, such non-neutralizing antibodies can contribute to the highly undesirable ADE (antibody-dependent enhancement) effect [11,26,27]. Thus, it is highly important to control the conformation of the S-trimers in the final vaccine product.

The structural and biochemical data published earlier for native SARS-CoV-2 were the starting point for our research [10,11,12,13,14,15]. For reasons of biosafety, we ourselves did not have the opportunity to investigate a non-inactivated virus using special equipment. We tested the efficiency of inactivation protocols in the BSL-3 laboratory and then transferred only fully inactivated preparations for further research to other laboratories outside BSL-3. The first task was to characterize the morphology of S-trimers on the surface of chemically or physically inactivated SARS-CoV-2 virions using transmission electron microscopy (TEM). We were interested in elucidating whether the spikes on the surface of virions inactivated by various agents are in a pre-fusion or post-fusion state. In addition, we aimed to characterize the protein composition of potential vaccine preparations and qualitatively evaluate their immunoreactivity (antigen affinity) via ELISA using two panels of sera collected from convalescents or people vaccinated by Sputnik V against naïve people. The express methods we used for sample preparation and investigation are suitable outside BSL-3 laboratories and could be applied during the downstream manufacturing process of inactivated SARS-CoV-2 vaccines.

## 2. Materials and Methods

### 2.1. Virus Growing and Inactivation

#### 2.1.1. Virus and Cells

The SARS-CoV-2 hCoV-19/Wuhan/WIV04/2019 strain was used in this work. Virus passaging was performed in Vero cells (ATCC, CCL81.4) in 1× MEM Alpha Medium (ά MEM, Thermo FS) supplemented with 10% fetal bovine serum (Thermo FS), 100 μg/mL gentamicin (Dalchimpharm, Khabarovsk, Russia), and 10 µg/mL amphotericin B (BioloT, Saint Petersburg, Russia). After the cellular monolayer was formed in a cell culture flask, the serum-supplemented MEM was replaced with a supporting medium (SM) containing 0.5% fetal bovine serum, 100 μg/mL gentamicin, and 5 µg/mL amphotericin B. The cells were infected with the virus in an SM (1 × 10^8^ TCID_50_/mL), adding it to the virus-free SM (0.6%, *v*/*v*). Passaging of SARS-CoV-2 in Vero cells was carried out at 37 °C in an atmosphere of 5% CO_2_. On the third day, a 100% cytopathic effect (CPE) was observed. After six subsequent passages, a medium containing the virus was collected, frozen, and thawed three times and clarified using centrifugation for 30 min at 1500–2000 rpm (Eppendorf 5920 R, S-4 × 1000). Aliquots of clarified virus-containing medium were frozen and kept at −80 °C until further usage. The combined virus harvest with 1 × 10^8^ TCID_50_/mL titer was used in all variants of inactivation. All work with the virus was performed in the BSL-3 environment.

#### 2.1.2. Infectious Activity Determination

The infectious activity of SARS-CoV-2 was determined by estimating the CPE on Vero cells in flat-bottomed 96-well plates (Eppendorf, Hamburg, Germany). A series of ten-fold dilutions of the initial virus-containing medium in an SM (1 × 10^1^ TCID_50_/mL to 1 × 10^7^ TCID_50_/mL) was prepared. Then, 200 μL of the prepared dilutions was added to each of the seven wells with a monolayer of Vero cells after removing from them the serum-supplemented MEM. At least 12 wells were left for cell culture control. The plates were incubated at 37 °C in an atmosphere of 5% CO_2_ until the appearance of a characteristic CPE, which was estimated on day 4 using light microscopy. The infectious virus titer (TCID_50_/mL) was determined using the Reed–Muench method [28].

#### 2.1.3. Virus Inactivation

The variants of virus inactivation are summarized in Table 1.

For chemical inactivation of SARS-CoV-2, β-propiolactone (BPL, Sigma–Aldrich, St. Louis, MA, USA) or paraformaldehyde (Sigma–Aldrich) were used. To obtain the BPL variant (1:1000), 0.2 mL of a freshly prepared 1% solution of BPL in an SM was added to 1.8 mL of the thawed (at 4 °C) aliquot of virus-containing medium and kept for 2 h at 4–8 °C. Similarly, to obtain the BPL variant (1:2000), 0.1 mL of a freshly prepared 1% solution of BPL in an SM was added to 1.9 mL of the virus-containing medium and kept for 36 h at 4–8 °C. After inactivation, 2 mL of virus-containing medium was added to tubes with 13 mL of cooled SM. The inactivating agent was removed using centrifugation in Amicon Ultra-15 100 kDa units at 3000 rpm for 3.5 min (Eppendorf 5920 R, S-4 × 1000), and the fresh SM was added to the virus. The procedure was repeated in triplicate. After the removal of the inactivating agent, small aliquots were taken to determine the infectious activity. Samples inactivated with paraformaldehyde were prepared in a similar way.

For physical inactivation, UV radiation at 205 nm was applied. The virus-containing samples were thawed at 4 °C for 4 h, placed in 6-well plates with 2 mL in each well. Different plates were used for each UV exposition time. The plates with samples were placed perpendicularly to the radiation source—the 30 W UV lamp (F30T8)—at a distance of 10 cm. Riboflavin (Pharmstandard, Dolgoprudny, Russia) was added from a stock solution (10 mg/mL) to half of the samples to a final concentration of 500 µM. After turning on the UV lamp, the plates were sequentially removed from the zone of UV radiation after 30 s, 1 min, 3 min, and 10 min. The accumulated intensity of light emitted from the UV light source was determined in the range of 200–320 nm using IMO-2N equipment (Etalon, Moscow, Russia) with a detector diameter of 15 mm. The resulting total radiation doses were 0.027 J/cm^2^, 0.047 J/cm^2^, 0.142 J/cm^2^, and 0.474 J/cm^2^, respectively. From each sample, virus-containing medium was taken to determine the infectious activity, and the remaining virus-containing medium was frozen and stored at −80 °C.

#### 2.1.4. Virus Concentration and Purification

The thawed virus samples were pelleted using centrifugation through a 20% (*w*/*v*) sucrose cushion in standard phosphate-buffered saline (PBS, pH = 7.4) using a Beckman TL100 centrifuge (TLS-55 rotor) at 40,000 rpm for 1.5 h at 4 °C. The virus pellets were resuspended in minimal volumes (30–50 µL) of PBS (pH 7.4) for further analysis.

### 2.2. Electrophoretic Analysis and Matrix-Assisted Laser Desorption/Ionization Time-of-Flight Mass Spectrometry (MALDI-TOF-MS) in-Gel Protein Identification

Concentrated virus samples were loaded onto 4–15% gradient gels (Bio-Rad) and run in Bio-Rad PROTEAN III equipment using Laemmli SDS-PAGE (sodium dodecyl sulfate polyacrylamide gel electrophoresis) [29] buffers. Coomassie G250 stained protein bands were developed as described previously [30]. Gel pieces of approximately 3 mm^3^ were destained with a 50 mM ammonium bicarbonate/40% acetonitrile solution, dehydrated with 100 μL acetonitrile, and rehydrated with 5 μL of digestion solution containing 20 mM ammonium bicarbonate and 15 ng/μL of sequencing-grade trypsin (Promega). Digestion was carried out at 37 °C for 5 h. Peptides were extracted with 10 μL of a 0.5% trifluoroacetic acid solution. To obtain the peptide mass fingerprint, 1 μL of the extract was mixed with 0.5 μL of a 2,5-dihydroxybenzoic acid-saturated solution in a 30% acetonitrile and 0.5% trifluoroacetic acid solution on a stainless steel MALDI sample target plate. Mass spectra were recorded on an UltrafleXtreme MALDI-TOF-TOF mass spectrometer (Bruker Daltonics, DE, Billerica, MA, USA) equipped with an Nd laser. The MH+ molecular ions were detected in reflecto-mode. The accuracy of the monoisotopic mass peak measurement was 50 ppm. Fragment ion spectra were obtained in lift mode. The accuracy of the fragment ion mass peak measurements was within 1 Da. Protein identification was carried out through MS + MS/MS ion searches using Mascot software (Matrix Science; http://www.matrixscience.com/ (accessed on 1 July 2022; 15 September 2022)) through the NCBI and home protein databases.

### 2.3. Transmission Electron Microscopy (TEM)

Virus samples were deposited on Formvar carbon-coated grids (TED Pella, Redding CA, USA) and incubated for two minutes. After that, the excess solution was removed, and the samples were then stained for 20 s with a 2% water solution of phosphotungstic acid (PTA; Sigma–Aldrich), pH 7.0. Samples were viewed with a transmission electron microscope Jeol JEM-2100 (JEOL Ltd., Tokyo, Japan) at an accelerating voltage of 200 kV, in parallel beam mode, with a defocus of 0.8 microns. Images were obtained and processed using a Gatan Orius SC200D (2 k × 2 k) detector and Gatan Digital Micrograph software (Gatan, Inc., Pleasanton, CA, USA).

### 2.4. Enzyme-Linked Immunosorbent Assay (ELISA)

The following variants of sera were used for the ELISA analysis: (1) convalescent sera, designated as R (12 samples); (2) archival sera collected in 2018 (before the pandemics), designated as naïve (10 samples); and (3) paired sera of people vaccinated with the Sputnik V vaccine on day 0 (d0) and day 42 (d42) (12 samples each). The paired sera samples were obtained from volunteers who gave their informed written consent. The project was approved by the ethics committee.

The total protein concentration within the inactivated preparations was measured using a Qubit fluorometer with supplied reagents (Thermo Fisher Scientific). All inactivated antigen preparations, at a concentration of 10 µg/mL, were adsorbed onto Nunc MaxiSorp plates (#442064) at +4 °C overnight. The antigen preparations were then removed, the plates were washed three times, and a blocking buffer containing 5% milk powder was added for 2 h at RT. Afterwards, all sera were diluted at a ratio of 1:100 in 5% milk powder and incubated with plates for 1 h at RT. Each serum was analyzed in duplicate. Anti-human IgG HRP (Sigma #A0170-1ML) was diluted at a ratio of 1:5000 and used as detecting antibodies, a TMB reagent (Biolegend 421101) was used as a substrate, and 2N H_2_SO_4_ was used as a stop solution. Absorbance was calculated by measuring an optical density (OD) at 450 nm (Multiscan SkyHigh, Thermo FS).

### 2.5. Statistical Analysis

All statistical analyses were performed using GraphPad Prism version 8.0.1 for Windows (GraphPad Software, San Diego, CA, USA). For each inactivated preparation measured against each group of sera (convalescent, grafted, and naïve), reactivity curves were built, and the area under the curve (AUC) was calculated, as well as the ratio of the AUC of convalescents to naive people and the people vaccinated with Sputnik V on day 42 to the same persons on day 0. The level of reliability of the increase in antibody affinity (OD in ELISA) in the sera of vaccinated people was calculated using Sidak’s multiple comparisons test. Neutralization antibody titers of all used sera were measured according to [31] and represented in Appendix A.

## 3. Results

### 3.1. Validation of Inactivation

In this work, we tested BPL, formaldehyde, and high-energy UV light, in some experiments with Riboflavin as an enhancer to inactivate SARS-CoV-2. The virus-containing medium with an initial 1 × 10^8^ TCID_50_/mL titer was used in all variants of inactivation. After chemical inactivation with BPL or formaldehyde, the infectious activity was measured for all tested variants (Table 1), and the total inactivation of the virus was observed (1 × 10^0^ TCID_50_/mL). To confirm the absence of a low concentration of the active virus, an additional passage in Vero cells was performed from the undiluted virus-containing medium revealing no CPE. Thus, all the selected chemical agents completely inactivated the SARS-CoV-2 virus in the virus-containing medium.

UV inactivation efficiency differed depending on the presence of riboflavin in the virus-containing medium (Appendix A). As shown in the graphs, starting from 1 min of UV irradiation, corresponding to 0.047 J/cm^2^, and at higher UV exposures to a virus-containing medium without riboflavin, no infectious activity was detected. However, none of the riboflavin-containing variants achieved complete virus inactivation in our experiments (Appendix A). Further, only UV-treated samples without a riboflavin addition were used in the work.

### 3.2. Various Inactivating Agents Affect the Virus Spikes’ Morphology Differently

Using TEM analysis, SARS-CoV-2 virions were found to have a spherical or slightly elongated morphology, with an average size of 80–120 nanometers, in all preparations of the inactivated virus-containing media. While examining virions, we noticed that many particles were entirely decorated with a typical S-spike “crown”, but some were partly spike-decorated or entirely spikeless. Earlier similar characteristics were observed for other coronaviruses [32].

Negative staining allowed us to unambiguously distinguish single spikes of different morphologies on the surface of virions that were earlier characterized by cryo-EM/ET and classified as flail-like spikes (wider ones, supposedly representing the pre-fusion conformation of the S-trimer) or needle-like spikes (narrower ones, supposedly representing the post-fusion conformation) [12,15]. We also found “rounded” spikes without a visible stem, probably resulting from a certain projection of the flail-like spike, which we classify at the moment as distorted/rounded flails. In addition to viral particles, the preparations contained impurities of the material of the destroyed cells. Typical micrographs obtained for the samples inactivated using formaldehyde, BPL, or UV radiation are presented below.

#### 3.2.1. Formaldehyde

In formaldehyde-inactivated samples (both 2% and 4%), the overwhelming majority of the spikes were in the flail-like or rounded flail-like conformation according to our classification (Figure 1A–D). There were almost no needle-like spikes found at the virions’ surface. Interestingly, sometimes flail-like spikes are seen at different angles to the viral membrane (see blue arrows in Figure 1B,D). High mobility of S-trimers was described earlier [10,12,13], suggesting several hinges in the thin stem of the spike, allowing it to deviate at 50–90 degrees from normal to the surface [15].

Cryo-EM analysis of the SARS-CoV-2 suspension fixed with 4% formaldehyde (30 min at room temperature) revealed 3% of spikes being in the post-fusion conformation and 97% in the pre-fusion conformation [10]. A similar proportion (4%) of needle-like spikes was found in our experiments (Table 2). Therefore, this type of inactivation mostly preserves the initial pre-fusion state of S-trimers.

#### 3.2.2. β-Propiolactone (BPL)

In the case of virus inactivation by BPL (1:2000, 36 h), two major conformations of S-spikes at the surface of SARS-CoV-2 virions were found: both the flails/rounded flails that we observed earlier in the formaldehyde-inactivated samples, reminiscent of the pre-fusion state of S-trimers (Figure 2A–C) and another conformation, seen as thinner and stiffer needles, supposedly being the post-fusion state of the spikes (Figure 2D–F). In virus preparations inactivated by a higher concentration of BPL during a shorter time period (1:1000, 2 h), there were also plenty of flail-like spikes observed (Appendix A) as well as a portion of needle-like ones (Appendix A). A simplified statistical analysis revealed 23% to 33% of needle-like spikes in the BPL-treated preparations (Table 2). It should also be noted that we observed more spikeless virions in the BPL (1:1000, 2 h) variant compared to the BPL (1:2000, 36 h) one, and the total number of calculated spikes in a population of about 50 virions turned out to be the smallest in the variant treated with a higher concentration of BPL compared with similar samplings of all other variants (Table 2).

#### 3.2.3. UV Irradiation

In the case of various regimes of UV radiation tested, we found flail-like spikes on the majority of virions (Figure 3A–C). Sometimes, needle-like spikes were also observed (Figure 3D; Table 2).

As can be seen from the micrographs presented above, the size of the populations of S-spikes having different morphologies (pre- or post-fusion conformations) varied depending on the method of virus inactivation. More often, the needle-like S-trimers, supposedly being in the post-fusion state, were found in the samples inactivated with BPL. Only small proportions of virions bearing such structures were found in the UV- and formaldehyde-inactivated samples. It should be emphasized that in the samples inactivated with BPL, only about 30% of the S-trimer population was needle-like, in contrast to Liu et al. [11], where more than two-thirds of the spikes were represented by needles. Unfortunately, for reasons of biosafety, we were unable to carry out a comparative analysis of native (non-inactivated) virus preparations to analyze the morphology of the S-trimers before the inactivation procedure.

### 3.3. Electrophoretic and Mass Spectrometric Analysis of Inactivated Virus Preparations

For SDS-PAGE analysis, we concentrated the virus using standard high-speed centrifugation protocols. Electrophoretic analysis of inactivated preparations showed that nucleoprotein (N) gives the best-seen protein band (~50 kDa) when the gels are dyed with Coomassie G-250 (Figure 4). Nucleoprotein is one of the coronavirus antigens [33,34]. Unlike N, S-bands are barely noticeable. This is not a surprise, since the number of copies (~15–180 according to different authors [10,12,13,15,34]) of S per SARS-CoV-2 virion is about ten times less than the N-protein of coronaviruses (~730–2200 copies per virion [32,34]). However, in-gel trypsin hydrolysis of protein bands and subsequent MALDI-TOF-MS analysis of the eluted peptide mixtures allowed us to identify the precursor S0 protein (non-processed by furin, ~175 kDa), as well as its cleavage products, S1/S2 protein bands (~100 kDa). It is noteworthy that S is highly N-glycosylated [35], which leads to a decrease in the number of identified peptides via a Mascot search (Appendix A). We also detected an M-protein (mixed with fragments of N) in several weak bands in a range of ~20–30 kDa (Figure 4 and Appendix A). The M glycoprotein is considered to be the most abundant constituent of coronaviruses [32,34] (~2000 copies). Despite this, it is poorly visualized using Coomassie blue staining [12]. As in the case of S, N-glycosylation of M [36] may reduce sequence coverage (Appendix A).

Earlier, Ke at al. [10] found that in released virions, S is present in both cleaved (S2, 73%) and uncleaved forms (S0, 27%). Our data allow us to assume that there is much more of the unsplit precursor S0, which may be due either to certain conditions of growing the virus or to specific protocols for preparing the sample for subsequent analysis. Thus, the procedure of virus concentration after thawing of frozen aliquots of virus-containing media resulted in significant losses of S. Apparently, a thin-hinged stem of the S-trimer [13] is a fragile structure that is being destroyed during high–speed centrifugation, especially after the freeze–thaw cycle.

Together with viral proteins, some concomitant proteins were detected in the virus preparations purified through a sucrose cushion, including albumin, obviously originated from the cell culture medium (in a greater proportion in the UV-inactivated sample), as well as chaperones, histones (Figure 4), and, in smaller quantities, other cellular proteins identified using Mascot searches. Thus, the inactivated virus should be purified more thoroughly for potential vaccine usage since concomitant proteins can cause undesirable allergic reactions.

### 3.4. Immunoreactivity of Inactivated Virus Preparations

The relative antigen affinity, or immunoreactivity [37], of virus preparations inactivated by different agents was measured via ELISA against several groups of human sera samples. After the raw ELISA data were collected (Figure 5), we calculated the average levels of immunoreactivity for each variant of virus inactivation measured against (i) convalescent relative to naïve sera collected before the COVID-19 pandemic or (ii) paired sera of people vaccinated with Sputnik V as areas under the curve (AUCs) (Table 3).

As can be seen from Table 3, the AUC values obtained for virus preparations inactivated with different agents and using different protocols vary significantly. Without taking into account the “background” (the reactivity of sera collected from non-immunized persons), the highest AUC value, both for the sera of convalescents and the sera of people vaccinated with Sputnik V, was shown to be the BPL-inactivated preparation (1:2000, 36 h). However, the ratio of the AUC of convalescents to the AUC of the sera of naïve people obviously more correctly reflects the immunoreactivity level. Regarding the sera of convalescents, UV- and formaldehyde-inactivated preparations had the highest activity, whilst against the sera of people grafted with the Sputnik V vaccine, BPL (1:2000, 36 h) and UV variants were most reactive. In addition, the BPL (1:2000, 36 h), UV, and formaldehyde variants demonstrated a reliable increase in antibody affinity in the paired sera test on day 42 after vaccination with Sputnik V (Appendix A). The least activity against both serum panels in all tests was demonstrated by the BPL (1:1000, 2 h) variant.

Interestingly, the R/naïve ratios were two to five times higher compared to d42/d0 ratios for all inactivated preparations. This implies a significant contribution to the interaction with antigens in the sera of convalescents not only of antibodies against S-protein, the main surface antigen, but also antibodies against N- and M-proteins, while only antibodies to S are present in the sera of people vaccinated with Sputnik V.

## 4. Discussion

Given the need for a sufficient amount of effective COVID vaccines, as the pandemic is still ongoing, the manageable control of conformational changes in the main SARS-CoV-2 S-antigen during the downstream manufacturing process of inactivated vaccines is of paramount importance. It is widely believed that the S-protein’s pre-fusion state used in all types of vaccines is a target conformation that elicits an optimal immune response. On the contrary, the post-fusion conformation of S can reduce the efficiency of vaccines, since it lacks the main epitopes stimulating the production of neutralizing antibodies. In this work, having in mind the necessity of developing an effective inactivated COVID vaccine protocol, we have tested several agents and protocols of SARS-CoV-2 inactivation, widely used in research, development, and manufacturing of inactivated vaccines. To achieve the necessary depth of the analysis, we applied the following analytical methods: TEM, a robust and reliable method to visualize the S-trimers’ shape; MALDI-TOF-MS to characterize protein composition; and ELISA to assess the immunoreactivity of inactivated virus preparations.

We have found that all tested protocols completely inactivated SARS-CoV-2 in the virus-containing medium, except for the UV variant in the presence of riboflavin. Previously, riboflavin was used as a photosensitizer in [38]. Goodrich et al. demonstrated that usage of riboflavin in the Mirasol^®^ Pathogen Reduction Technology System resulted in a decrease in the TCID_50_/_mL_ titer of various viruses in physiological fluids by 10^4^–10^6^ times [39], which only partially coincides with our data. Perhaps riboflavin absorbs part of the UV-radiation spectrum. In the absence of riboflavin, our results were the same as shown earlier—UV exposure of 0.025 J/cm^2^ [16], 0.04 J/cm^2^ [40], or higher, caused a complete inactivation of SARS-CoV-2.

TEM image analysis revealed that, in preparations inactivated with BPL, there were large proportions of S-spikes in the post-fusion conformation, while just a small proportion of post-fusion spikes were observed in the formaldehyde and UV variants. Earlier, several groups of authors applied TEM to give a glimpse of the morphology of SARS-CoV-2 S-spikes at the surface of virions inactivated by BPL [16,22,23,24], and some investigated the S-trimer’s architecture in more detail using cryo-EM and cryo-ET [11]. The results differed drastically. While Liu et al. [11] found the majority (74%) of spikes to be in the post-fusion conformation after virus treatment with 0.05% BPL (1:2000) for 36 h at 4 °C, with following isopycnic ultracentrifugation to concentrate the virus, Yu et al. [24] did not report obvious post-fusion S-spikes in preparations treated with a more diluted solution of BPL (1:4000) for a shorter period of time (20–24 h, 2–8 °C) and further pelleting of the virus through 300 kDa filters at low speed. In our experiments, one of the inactivation protocols (BPL 1:2000, 36 h) was identical to Liu’s et al. [11], but it differed in post-inactivation procedures; namely, we did not apply ultracentrifugation to concentrate the virus before the TEM analysis that may help to retain the native state of the spikes [10]. Obviously, various factors, including virus strains, may affect spike morphology.

Using mass spectrometry, we confirmed the presence of all major structural proteins (S, N, and M) of SARS-CoV-2 in the preparations, albeit in lower amounts than expected. That may be due to a partial loss of a fragile S-protein during inactivation and/or concentration procedures coupled with a freezing–thawing cycle and the ultracentrifugation that follows.

Employing ELISA for inactivated virus preparations, we first calculated the ratio of reactivity of convalescent sera possessing the wide repertoire of SARS-CoV-2 antibodies to naïve human sera collected before the pandemic. In this case, the least immunoreactivity was demonstrated in the BPL-inactivated samples, and the higher the concentration of the inactivating agent, the worse the detected response. This might be collectively due to the presence of a sufficient number of antibodies in convalescents recognizing various epitopes of all virus proteins, and to destructive effects, especially from the BPL (1:1000, 2 h) treatment.

In addition, we have estimated with ELISA the immunoreactivity of inactivated virus preparations against paired sera collected from people vaccinated with Sputnik V, the gene-engineered vectored type of vaccine. This vaccine, when entering the cell, triggers the synthesis of S-protein in the pre-fusion conformation. Thus, we could only evaluate the immunoreactivity of the S-antigen against the S-protein’s antibodies. The BPL (1:2000, 36 h) variant demonstrated 1.1–1.7 times better reactivity compared to the UV and formaldehyde variants, respectively, while the BPL (1:1000, 2 h) variant again demonstrated the least activity.

Interestingly, the N- and M-proteins present in the inactivated virus samples in addition to the S-antigen increased the absorption signal several times in the ELISA performed with the convalescent sera compared to the sera of people vaccinated with the Sputnik V vaccine (Figure 5; Table 3).

It should be stressed that the “harsher” BPL variant (1:1000, 2 h) is about 2–3 times less active compared to three other variants tested when measured against the human-paired sera of people vaccinated with Sputnik V, and about 4–5.5 times less active compared to three other variants when measured against the sera of convalescents (see the R/naïve and d42/d0 ratios in Table 3). In fact, this is the only “bad” protocol, seriously losing to three other variants tested.

The difference observed in ELISA between the two used BPL protocols—(1:2000, 36 h) and (1:1000, 2 h)—can hardly be explained by the overall architecture of S-trimers only, since there was no dramatic difference observed between them in the amount of needle-like spikes, according to our TEM analysis. We may hypothesize that there is a more subtle difference between S-proteins in these two BPL variants, not visible using TEM and hardly visible using cryo-electron microscopy, namely, a difference in the degree of their chemical modifications. It is shown that BPL may not only modify nucleic acids but is also capable of interacting with various amino acids, yielding alkylated and acylated products and undesired long chains [19,41]. Thus, plenty of BPL-modified peptides were detected using mass spectrometry on the surface of influenza virus glycoproteins hemagglutinin and neuraminidase [42]. Obviously, such modifications may mask antigenic epitopes and thus decrease the immunogenicity of protein antigens, and a higher concentration of BPL may cause a more drastic effect. To clarify this hypothesis, further investigations in this field would be interesting. In addition, the increase in the number of particles devoid of spikes observed in the population of virions treated with a high concentration of BPL is also important, since, naturally, virus particles without spikes or having few spikes should be less immunoreactive compared to particles with S-spikes.

To conclude, TEM analysis of negatively stained virions seems to be a helpful method for manufacturer’s quality assurance/quality control (QA/QC) which is a key factor for COVID-19-inactivated vaccine effectiveness. For a more detailed study of the structure of spikes on the surface of inactivated virions and for a more reliable quantification of the ratio of different conformations of spikes in samples, it is necessary to use high-resolution microscopy methods such as cryo-EM and cryo-ET, as well as algorithms for automatic image analysis [43]. However, such an analysis is expensive, complex, and hardly suitable for controlling the vaccine production process. On the other hand, our approach, combining different but rather routine methods such as TEM analysis of negatively stained viral spikes, mass spectrometry-based characterization of the preparations, and their testing with ELISA using the sera of convalescents and vaccinated people helps one eliminate some of the inactivation conditions that are not appropriate for testing their immunogenicity in an animal model.

A future study of an inactivated virus using the ELISA method, if it becomes possible, may add interesting information about the immunoreactivity of native viral preparations. This will help to clearly distinguish which affinity properties the native SARS-CoV-2 has, and what changes result from exposure to an inactivating agent. However, based on our comparative immunoreactivity study of several inactivated preparations, we conclude that the BPL (1:1000, 2 h) variant, which is routine during the manufacturing process of inactivated influenza vaccines, is not good enough to be proposed for inactivated COVID-19 vaccine production because of some peculiarities of the spike antigen.

## Figures and Tables

**Figure 1 viruses-15-00480-f001:**
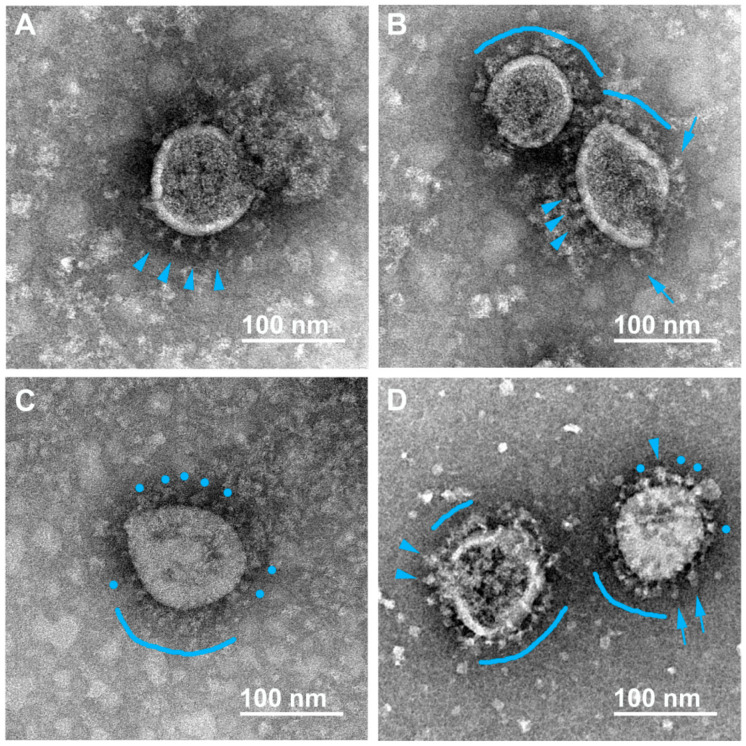
Electron microscopy images of SARS-CoV-2 virions inactivated with 2% formaldehyde (1 h, 37 °C). Various forms of flail-like spikes supposedly corresponding to pre-fusion conformation are observed (**A**–**D**); single “classical” flails (**A**,**B**,**D**) and distorted/rounded flails (**C**,**D**) are indicated with blue arrowheads and blue circles, respectively. The flail-like spikes essentially tilted to the viral membrane are indicated with blue arrows (**B**,**D**). Coats of flail-like spikes are marked with blue arches (**B**–**D**).

**Figure 2 viruses-15-00480-f002:**
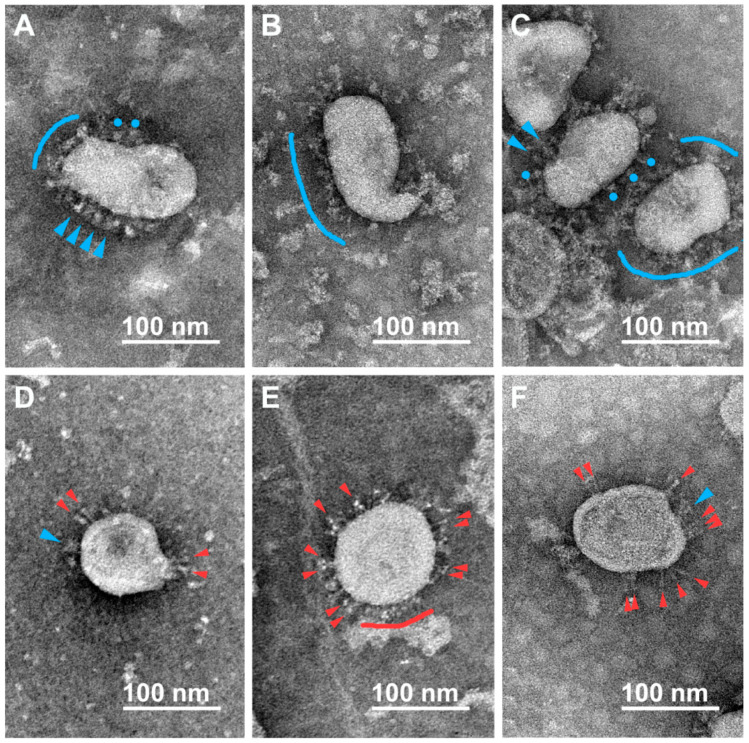
Electron microscopy images of SARS-CoV-2 virions subjected to inactivation with BPL (1:2000, 36 h, 4–8 °C). Single spikes of different morphologies are distinguishable on the virions’ surfaces. Flails (blue arrowheads), rounded flails (blue circles), and flail coats (blue arches) supposedly correspond to the pre-fusion conformation of S-trimers (**A**–**C**), while needles (red arrowheads) and their coats (red arches) supposedly correspond to the post-fusion conformation of S-trimers (**D**–**F**).

**Figure 3 viruses-15-00480-f003:**
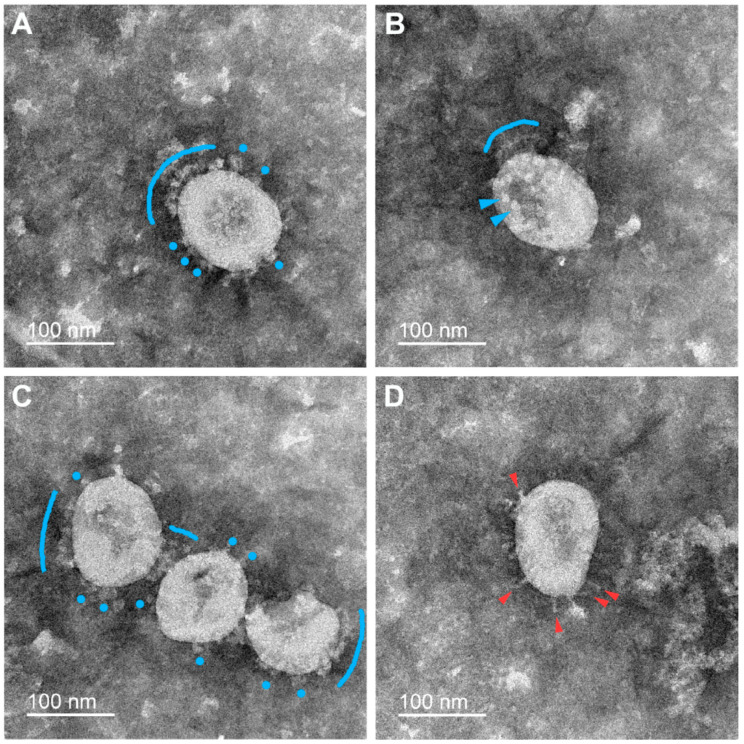
Electron microscopy images of SARS-CoV-2 virions subjected to UV inactivation (1 min). The flail-like (supposedly pre-fusion) spikes are marked in (**A**–**C**): flails (blue arrowheads), rounded flails (blue circles), and flail coats (blue arches). The needle-like (supposedly post-fusion) spikes are rarely observed on some virions (**D**) (red arrowheads).

**Figure 4 viruses-15-00480-f004:**
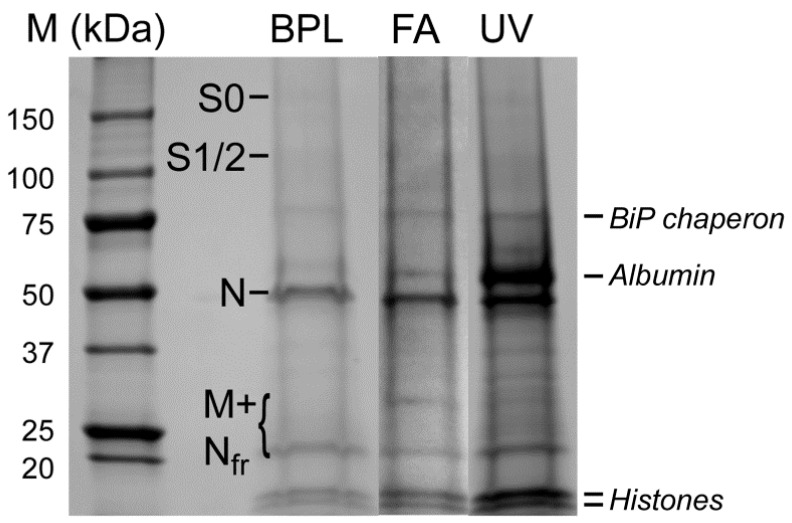
Electrophoregram of preparations of viral particles inactivated by BPL (1:2000 for 36 h), formaldehyde (FA, 2%), or UV-radiation (UV, 3 min). Prior to loading onto the gel, the samples were concentrated via ultracentrifugation through a 20% sucrose cushion. Shown are the positions of SARS-CoV-2 structural proteins: S0—the non-split precursor form of S; S1/S2—cleavage products of S; M—membrane protein; N—nucleoprotein; Nfr—nucleoprotein’s fragments. Some admixed proteins are also indicated: BiP chaperon, albumin, and histones. All proteins were identified using Mascot searches after in-gel trypsin hydrolysis of protein bands, elution of peptides, and their MALDI-TOF-MS analysis.

**Figure 5 viruses-15-00480-f005:**
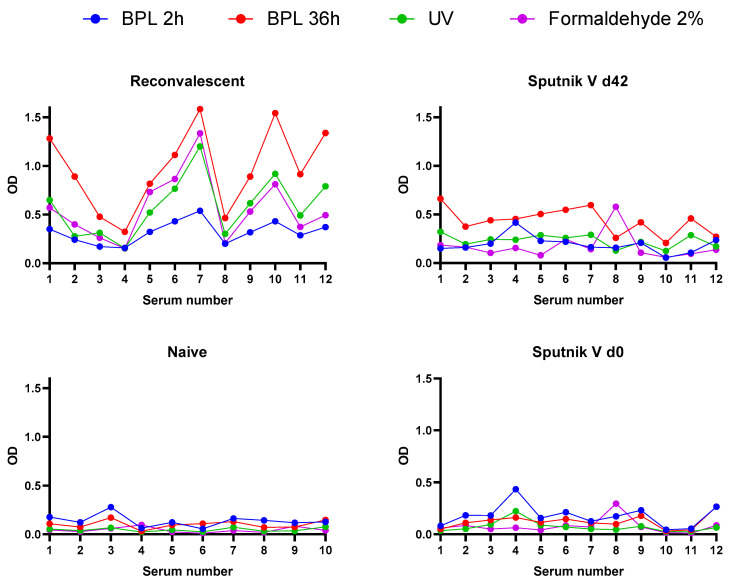
Immunoreactivity curves of inactivated virus preparations measured against sera of convalescent/naïve persons and paired sera of people vaccinated with Sputnik V. The absorbance data (OD 450) were plotted against the respective serum designated by a number; d0—the serum collected before the vaccination; d42—the serum collected on the 42nd day after vaccination; BPL 2 h—β-propiolactone (1:1000, 2 h); BPL 36 h—β-propiolactone (1:2000, 36 h).

**Table 1 viruses-15-00480-t001:** Variants of virus inactivation.

№	Inactivation	Inactivating Agent	Inactivation Conditions	Temperature
Chemical	Physical
1	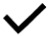		BPL, 1:1000	2 h	+4–8 °C
2	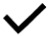		BPL, 1:2000	36 h	+4–8 °C
3	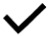		formaldehyde, 2%	1 h	+37 °C
4	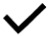		formaldehyde, 4%	1 h	+37 °C
5		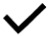	UV + Riboflavin	0.027–0.474 J/cm^2^	RT
6		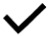	UV	0.027–0.474 J/cm^2^	RT

**Table 2 viruses-15-00480-t002:** Morphology of S-spikes at the surface of inactivated SARS-CoV-2 virions.

Spike Morphology/Inactivated Sample	Number of Virions	Number of Spikes *	Shape of the Spike, Number (%)		
			Flail-Like	Needle-Like	Unidentified
Formaldehyde (2%)	44	405	381 (94)	15 (4)	9 (2)
UV (1 min)	52	595	541 (91)	31 (5)	23 (4)
BPL (1:2000; 36 h)	47	346	246 (71)	78 (23)	22 (6)
BPL (1:1000; 2 h)	51	229	135 (59)	75 (33)	19 (8)

* Single spikes around the virion, but no dense spike coats were taken into consideration.

**Table 3 viruses-15-00480-t003:** Immunoreactivity of SARS-CoV-2 inactivated with different agents measured with ELISA.

Inactivated Sample/Sera *	Area under Curve (AUC) **			
	Formaldehyde (2%)	BPL (1:2000, 36 h)	BPL (1:1000, 2 h)	UV (3 min)
R	6.204	10.33	3.464	6.282
Sputnik V d42	1.897	4.735	2.117	2.523
Naïve	0.41	0.889	1.225	0.4055
Sputnik V d0	0.863	1.292	1.972	0.803
R/Naïve	15.13	11.62	2.83	15.49
d42/d0	2.20	3.66	1.07	3.14

* R—convalescent; naïve—collected in 2018; d42 and d0—collected on day 0 and day 42 after vaccination with Sputnik V; ** calculated based on the graphics represented in Figure 5 using GraphPad software.

## Data Availability

Data is available upon reasonable request.

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
