# Peer review of "Structural and Immunoreactivity Properties of the SARS-CoV-2 Spike Protein upon the Development of an Inactivated Vaccine"

_viruses, 2023, doi:10.3390/v15020480_

Round 1

Reviewer 1 Report

Kordyukova et al. Investigate the impact of different inactivation methods on the structure and therefore the immunogenicity of the SARS-CoV-2 Spike (S) protein in the context of inactivated virions. The authors use TEM analysis to determine whether each method broadly presents S in one of two major conformations, the pre-fusion, ‘flail-like’ conformation or the post-fusion, ‘needle-like’ conformation. The authors show that formaldehyde inactivation preserved the pre-fusion conformation best, followed by UV inactivation with BPL inactivation performing worst in this assessment. The authors went on to analysis the UV and BPL inactivated samples using MALDI-TOF MS before assessing the immunogenicity of the different conditions against a number of different sera samples, including naïve patients, convalescent patients and sera from patients vaccinated with the virally vectored vaccine, Sputnik. Overall, the authors conclude that TEM analysis offers a good method for quality control for inactivated Covid-19 vaccines. This study is well-executed and would be of interest to the readership, with the revisions suggested below, I believe this manuscript will soon be ready for publication.

Major Comments

Figure 1, 2 & 3 – Would it be possible to include a wider field of view (lower magnification) of the TEM and then show the high magnification images in Figure 1

Figures 1, 2 & 3 – I feel it would significantly improve the manuscript if there were further quantitative analysis carried out on the TEM images obtained for all three figures. For example, it should be possible for the authors to take a number of images per condition, and analysis 50 – 100 particles using 2D classification to determine the proportion of ‘Flail-like’ and ‘Needle-like’ Spikes in each condition, allowing the readership to see a larger pool of data than the select images in these figures.

Figure 4 – Is it possible to compare these samples to a sample of non-inactivated virus to see how they compare?

Table 2 – Is not completely intuitive to the reader, would it be possible for the authors to include the graphical data for this table in the main results rather than in the supplementary material?

Minor Comments

Line 37 – Typo to remove ‘The introduction’ from the beginning of the sentence.

Line 52 – ‘Is an S-protein’ should read as ‘is the S-protein’

Line 152-153 – Font mismatch

The authors refer to 8 lg TCID50/mL throughout the manuscript, however this is not a common way to refer to TCID50 measurements. Could the authors please alter this to read for example 1 x 107 TCID50/mL for ease of understanding for the readership.

Unsure if this is a journal requirement but it would be good to see Table 1 near where it is first mentioned in the results section rather than in the methods.

Line 226 – This sentence feels slightly off grammatically I would suggest it should read ‘SARS-CoV-2 virions were found to be spherical or slightly elongated morphology’

Figure 1 & 2 – Please ensure each micrograph presented provides a scale bar as in Figure 3.

Line 284 – I suggest the authors replace ‘mircophotographs’ with ‘micrographs’

Reviewer 2 Report

Thanks for the opportunity to review this paper.

This paper investigated three inactivated methods (by β-propiolactone, formaldehyde, and UV-radiation) for vaccine development and focused on their influences on the morphology and immunoreactivity of SARS-CoV-2 spike protein. The authors applied the inactivated viruses to TEM and observed the shape of spike protein displayed on the surface of the virus. They also confirmed the composition of the samples by MALDI-TOF-MS. The authors also checked the immunoreactivity of the inactivated virus sample through ELISA with several human sera samples. However, similar research has been well studied previously, which greatly reduced the novelty of this paper. The inactivated methods used here were commonly used and the TEM analysis for viruses was also not new. In addition, some other issues also need to be addressed.

1. Both TEM analysis and the immunoreactivity test against human sera need a positive control of SARS-CoV-2 virus without treatment.

2. Figure 4 only showed BPL and UV treatments, wild-type and formaldehyde treated virus samples should be included. As shown in this figure, the proteins contained in BPL and UV treated samples are totally different in amount. How did the authors normalize the samples for the immunoreactivity testing?

3. The TEM images shown in Figure 1-3 presented some images for the viruses with different inactivated treatments. They all showed a wide range distribution of the spike protein conformations. As the images only showed several viruses, could the authors provide a statistical analysis of each conformation for each sample.

4. The authors concluded that "Formaldehyde and UV-inactivated samples demonstrated the highest affinity/ immunoreactivity against the convalescent sera, while β-propiolactone (1:2000, 36h) and UV-inactivated ones were more active against the sera of people vaccinated with Sputnik V." However, in table 2, against the convalescent sera, AUC value of β-propiolactone (1:2000, 36h) is 10.33 which is higher than Formaldehyde (6.204) and UV (6.282); against the sera of people vaccinated with Sputnik V, the β-propiolactone (1:1000, 2h) AUC is 2.117 which is similar to UV (2.523). Is not it? The ratios comparison between R/Naïve and d42/d0 is not convinced as d0 should be similar to Naïve if not infected by SARS-CoV-2. 

5. Line 286, From the TEM images, "More often, the needle-like S trimers supposedly being in the post-fusion state were found in the samples inactivated with BPL." However, in table 2, the BPL treated samples have higher AUC value. How to explain this?

Round 2

Reviewer 1 Report

I appreciate the efforts taken to try and improve this manuscript and would like to congratulate the authors on their manuscript

Author Response

The authors thank the reviewer for his kind attitude towards our work and for all suggestions and corrections during the reviewing process.

Reviewer 2 Report

The authors addressed most of my concerns. However, the conclusions are not convincing due to the missing control group of wild type non-inactivated SARS-CoV-2. It is not clear how the inactivated treatments change the conformation of spike protein and how this conformation change influences the immunoreactivity.
